# Genotypes and phylogenetic analysis of adenovirus in children with respiratory infection in Buenos Aires, Argentina (2000–2018)

**Débora N. Marcone**[1,2,3◉], **Andrés C. A. Culasso**[2,3◉], **Noelia Reyes**[1], **Adriana Kajon**[4], **Diana Viale**[5], **Rodolfo H. Campos**[2,3], **Guadalupe Carballal**[1,3], **Marcela Echavarria**[1,3]*

**1** Unidad de Virología, Centro de Educación Médica e Investigaciones Clínicas (CEMIC) Hospital Universitario, Ciudad de Buenos Aires, Argentina, **2** Departamento de Microbiología, Inmunología y Biotecnología, Cátedra de Virología, Facultad de Farmacia y Bioquímica, Universidad de Buenos Aires, Buenos Aires, Argentina, **3** Consejo Nacional de Investigaciones Científicas y Técnicas (CONICET), Buenos Aires, Argentina, **4** Infectious Disease Program, Lovelace Respiratory Research Institute, Albuquerque, NM, United States of America, **5** Departamento de Microbiología, Hospital Prof. Dr. Juan P. Garrahan, Ciudad de Buenos Aires, Argentina

◉ These authors contributed equally to this work.
* mechavarria@cemic.edu.ar

**Data Availability Statement:** Sequence data are available in Genbank under accessions numbers: MG000707 to MG000784 and MK913783 to

## Abstract

Human adenoviruses (HAdV) are one of the most frequent causes of respiratory infections around the world, causing mild to severe disease. In Argentina, many studies focused on the association of HAdV respiratory infection with severe disease and fatal outcomes leading to the discovery in 1984 of a genomic variant 7h associated with high fatality. Although several molecular studies reported the presence of at least 4 HAdV species (B, C, D and E) in Argentina, few sequences were available in the databases. In this study, sequences from the hexon gene region were obtained from 141 patients as a first approach to assess the genetic diversity of HAdVs circulating in Buenos Aires, Argentina. Phylogenetic analysis of these sequences and others recovered from public databases confirmed the circulation of the four above-mentioned species represented by 11 genotypes, with predominance in species B and C and shifts in their proportion in the studied period (2000 to 2018). The variants detected in Argentina, for most of the genotypes, were similar to those already described in other countries. However, uncommon lineages belonging to genotypes C2, C5 and E4 were detected, which might indicate the circulation of local variants and will deserve further studies of whole-genome sequences.

## Introduction

Human adenoviruses (HAdV) are one of the most frequent causes of acute respiratory infections (ARI) around the world. The symptoms of HAdV infections can range from mild to severe even leading to long-term lung sequelae or death. Common adenoviral infections

MK913843. All relevant data are within the manuscript and its Supporting Information files. Additional methods and data are available in Mendeley Data: http://dx.doi.org/10.17632/jnh663g7w9.1 and http://dx.doi.org/10.17632/prf5vffgd2.1.

**Funding:** This work was funded by the ANPCyT-Argentina (PICT 2006-650, given to Dr. Echavarria) and by the Unit of Research-Fundación Allende (Young Investigator 2014, given to Dr. Marcone).

**Competing interests:** The authors have declared that no competing interests exist.

include respiratory, ocular and gastrointestinal manifestations. Less frequent manifestations include hepatitis, hemorrhagic cystitis, meningitis, and disseminated disease mostly described in immunocompromised patients [1]. Wide variation in clinical manifestations are attributed to the broad tissue spectrum tropism exhibited by HAdV various species and serotypes and also to differences in the host immune status [1]. The incidence of HAdV respiratory infections in children ranges from 2–20% depending on the studied population and detection method [2–5]. In Argentina, many studies focused on the association of HAdV respiratory infection with severe disease and fatal outcomes. Furthermore, the genomic variant designated 7h, associated with high fatality, was described for the first time in Argentina in 1984 [6, 7] and was subsequently reported worldwide [6, 8–10].

HAdV are classified into 7 species (A—G). Within each species, HAdV are sub classified into serotypes or genotypes. The initial 51 serotypes were determined by neutralization assays while genotypes 52 to 90 were described by bioinformatics analysis of whole-genome sequences [11, 12].

Historically, serotypes were determined by classical methods such as neutralization or complement fixation assays. Within one serotype, several genomic variants can be discriminated by restriction enzyme analysis (REA). The digestion profile of the prototype strain is designated as "p" while the other profiles are designated with letters "a", "b", etc. [13]. Currently, HAdV are mostly classified (or typed) into genotypes by sequence analysis. For typing purposes, the hexon gene is one of the most common targets to be amplified and sequenced but also the fiber, penton base, and polymerase genes are used [14, 15].

The most frequent genotypes associated with pediatric respiratory infection are 1, 2, 5 (species C), and genotypes 3 and 7 (species B). Genotype 4 (species E) is also associated with respiratory infections but mostly among military recruits and at a low frequency in the general population [1, 3, 4].

In Argentina, species C was the most frequently detected in outpatients from 6 to 48 months old with Flu-like symptoms during a study performed in 2002 [2]. In contrast, species B (72%) was more frequently detected than species C (26%) in hospitalized children with ARI in a cohort study conducted between 1999 and 2010 [16]. Despite the use of molecular methods in these studies, few sequences are available for phylogenetic analysis.

The aim of this study was to determine the circulating HAdV genotypes during a 14-year period in Buenos Aires, Argentina and to establish their phylogenetic relation to those described worldwide.

## Material and methods

### Study design

A cross-sectional study was performed using HAdV positive respiratory samples from children with ARI, in Buenos Aires, Argentina from 2000–2005, 2008–2011, 2014–2016, and 2018.

As part of a surveillance study of respiratory infections at CEMIC University Hospital, respiratory samples were collected from hospitalized children or outpatient children (attending the emergency room) with ARI from 2008 to 2018. In addition, samples from HAdV positive patients hospitalized with ARI at Prof. Juan P. Garrahan Children's Hospital from 2000 to 2005 were included.

Respiratory samples included nasopharyngeal aspirates from hospitalized children and nasopharyngeal swabs from outpatients. An informed consent signed by parents/legal-guardian was obtained for each patient and a specific form was filled with demographic data, underlying conditions, and clinical characteristics. Medical procedures during the hospital stay

including length of stay, oxygen therapy, admission to the intensive care unit, and mechanical ventilation support were recorded.

HAdV diagnosis was performed by rapid antigen detection by immunofluorescence assay (IFA) using specific monoclonal antibodies (Chemicon Millipore). An aliquot of the original frozen clinical specimen (-70˚C) was used for HAdV typing at CEMIC Clinical Virology Unit. Inoculation of positive samples was performed in cell lines for viral isolation and amplification for further REA analysis.

This study was approved by the Institutional Ethics and Review Committees of the Hospital Prof. Juan P. Garrahan and Hospital Universitario CEMIC (No. 466). IRB00001745-IORG0001315.

## HAdV phylogenetic analyses

Since HAdV sequences from different species are too divergent to be included in a reliable multiple sequence alignment, local alignments with the reference dataset using FASTA36 software [17] were used to initially classify species. Then, for both genotyping and phylogenetic diversity assessment, maximum likelihood and Bayesian methodology were employed. For genotyping, three datasets including all complete HAdV reference genomes were assembled: one for species B and E, another for species C and the last one for species D. For phylogenetic diversity assessment, nine datasets compatible with our analyzed region (see PCR and sequencing) were constructed. Each dataset included genotypes with similar hexon sequences: B3 and B68; B7 and B66; B11 and B55; B35; C1; C2; C5; D8 and E4. Sequence data source and analysis were graphically depicted in a flowchart (S1 Fig).

Predicted amino acid sequences were aligned with Muscle v 3.05 program [18] in SeaView v4 alignment viewer [19].

Maximum likelihood phylogenetic trees were obtained by a heuristic search as implemented in PhyML v3.0 [20]. The models of nucleotide substitution were selected according to the Bayesian Information Criterion implemented in jModelTest 2.16 [21]. Branch support was assessed by non-parametric bootstrap (1000 pseudoreplica). Genotype assignment was performed by clustering the HAdV study sequences with the reference (see Selection of HAdV reference sequences). Bootstrap values greater than 70% were considered significant evidence for phylogenetic grouping.

Bayesian trees were constructed with MrBayes v3.2.6 [22]. The number of substitutions (nst) and rate (heterogeneity in substitutions rates) were set according to the previously selected substitution model for each dataset. The Metropolis coupled Monte Carlo Markov chains were run for 5000000 of generations for all datasets. Chain convergence was assessed by visual inspection of the parameters trace files, the effective size number of each parameter (more than 200) and the comparison of two independent runs as featured in the MrBayes program.

The unrooted trees obtained were visualized, midpoint rooted, and converted into graphics with FigTree v 1.4.4 program (http://tree.bio.ed.ac.uk/software/figtree/).

**PCR and sequencing.** Viral DNA extraction was performed on an aliquot of the frozen clinical specimen with commercial columns (QIAamp DNA Mini Kit—QIAGEN), following manufacturers' protocol. Direct sequencing of the hypervariable regions 1 to 6 (HVR 1–6) of the hexon gene was used as a rapid molecular alternative for seroneutralization assays since it contains most of the immunologic relevant domains, as recommended by Lu & Erdman [15]. This region was amplified by PCR using the Hot StarTaq DNA Polymerase kit (QIAGEN) according to a previously published protocol [15]. The products of this nested PCR protocol ranged from 688 to 821 nucleotides in length depending on the HAdV species and genotype.

They were direct-sequenced by an automatic sequencer 3730XL by Macrogen (South Korea) after purification by ethanol precipitation.

Sequences were visually inspected, manually edited with BioEdit v7.0.5.3 [23], translated and aligned with ClustalW v1.81 [24]. HAdV sequences were deposited in GenBank under accession numbers: MG000707-MG000784 and MK913783-MK913843.

**Selection of HAdV reference sequences.** All available HAdV reference genotypes (n = 94) were included in the analysis for accurate hexon genotype identification (available at http://dx.doi.org/10.17632/prf5vffgd2.1). This reference dataset was created using two sources: the complete list of genomes displayed at web site https://sites.google.com/site/adenoseq/ (consulted on April 2020) and publications on HAdV molecular epidemiology [15, 25, 26].

**Genetic diversity assessment.** Phylogenetic analyses based on the HAdV partial hexon region (HVR 1–6) were performed with our sequences and those of the same genotype available in public databases. To obtain the most comprehensive number of HAdV sequences at the time of analysis, GenBank nucleotide searches and manually downloaded sequences from relevant bibliographic searches were included. The complete list of consulted bibliography is available at http://dx.doi.org/10.17632/jnh663g7w9.1.

## Viral culture and restriction enzyme analysis

Selected samples (n = 84) were inoculated in A549 cells until HAdV-cytopathic effect was observed and confirmed by IFA. HAdV isolates were reinoculated in A549 to obtain a higher viral concentration. The second passages were sent to the Lovelace Respiratory Research Institute (LRRI), U.S.A., for purification of intracellular viral DNA and REA analysis as previously described [27]. Restriction enzyme profiles obtained with a panel of endonucleases were analyzed by horizontal agarose gel electrophoresis and visualized by UV transillumination.

REA results were used to complete the characterization of isolates belonging to genotypes B66 (B7 hexon sequence but 7h genome-type [28]) and B68 (B3 hexon sequence but intertypic recombinant 3–16 genome [29]).

## Statistical analysis

A logistic regression analysis was performed to evaluate the risk of the clinical outcome severity of HAdV infection in association with the detected HAdV species and genotypes. For this purpose, children were classified in children who recovered, children who developed long-term lung sequelae, or children who died. Variables as age, sex, preterm born were introduced in the multivariate model. The first analysis included HAdV species B versus species C; and the second analysis included the most frequently detected HAdV genotypes. Statistical significance was assumed for p values <0.05. Statistical analysis was performed using STATA 12.0 (Stata Corp, College Station, TX).

## Results

### HAdV genotypes

A total of 148 HAdV positive samples from patients with ARI in Buenos Aires were selected to determine their HAdV genotypes. Among these, 141 (95.3%) were successfully typed by PCR amplification, sequencing, and phylogenetic analysis of HVR 1–6 of the hexon gene.

As a result of the local alignment analysis, samples were classified into four HAdV species: B (50.4%), C (40.4%), D (1.4%) and E (7.8%). The phylogenetic trees constructed for genotyping (Fig 1) allowed the identification of 11 genotypes, being B3 (27.0%) and C2 (20.6%) the

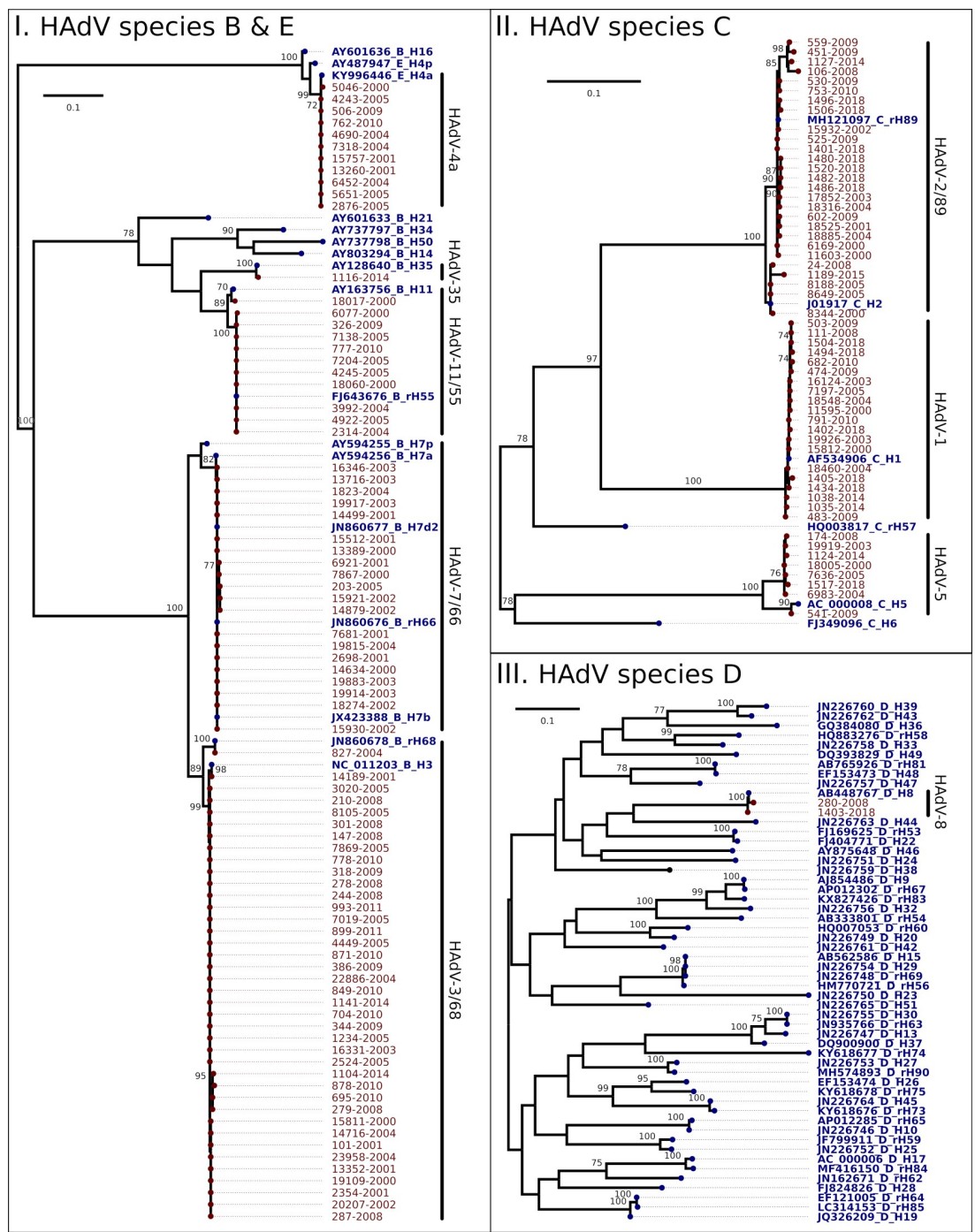

**Fig 1. Phylogenetic tree of HAdV strains from children with acute respiratory infection in Buenos Aires, Argentina -2000 to 2018.** Buenos Aires samples are in red, named by the laboratory identification number followed by the isolation year. GenBank accession numbers of all available reference sequences (blue) are shown. Phylogenetic trees obtained by Maximum Likelihood methodology based on the HAdV partial hexon gene (HVR 1–6) by HAdV species. For each dataset the best fit model was used: I. TPMuf+Γ, II. TIM2+ Γ+I, III. TIM2+ Γ+I. The branch support was determined by bootstrapping (1000 pseudoreplica).

most frequently detected, followed by B66 (14.2%), C1 (14.2%), E4 (7.8%), B55 (7.1%), C5 (5.7%), and in a lower frequency D8 (1.4%), B11 (0.7%), B35 (0.7%) and B68 (0.7%). Same results were obtained after the examination of the Bayesian version of the same phylogenetic

**Table 1.** HAdV genotypes detected by year in children with acute respiratory infection in Buenos Aires, Argentina from 2000 to 2018.

| Species | Genotype | 2000 | 2001 | 2002 | 2003 | 2004 | 2005 | 2006 | 2007 | 2008 | 2009 | 2010 | 2011 | 2012 | 2013 | 2014 | 2015 | 2016 | 2017 | 2018 |
|---|---|---|---|---|---|---|---|---|---|---|---|---|---|---|---|---|---|---|---|---|
| B | 3 | 2 | 3 | 2 | 1 | 3 | 7 | N/A | N/A | 7 | 3 | 6 | 2 | N/A | N/A | 2 | 0 | N/A | N/A | 0 |
| | 11 | 1 | 0 | 0 | 0 | 0 | 0 | N/A | N/A | 0 | 0 | 0 | 0 | N/A | N/A | 0 | 0 | N/A | N/A | 0 |
| | 35 | 0 | 0 | 0 | 0 | 0 | 0 | N/A | N/A | 0 | 0 | 0 | 0 | N/A | N/A | 1 | 0 | N/A | N/A | 0 |
| | 55 | 2 | 0 | 0 | 0 | 1 | 5 | N/A | N/A | 0 | 1 | 1 | 0 | N/A | N/A | 0 | 0 | N/A | N/A | 0 |
| | 66 | 3 | 4 | 4 | 5 | 2 | 2 | N/A | N/A | 0 | 0 | 0 | 0 | N/A | N/A | 0 | 0 | N/A | N/A | 0 |
| | 68 | 0 | 0 | 0 | 0 | 1 | 0 | N/A | N/A | 0 | 0 | 0 | 0 | N/A | N/A | 0 | 0 | N/A | N/A | 0 |
| C | 1 | 2 | 0 | 2 | 2 | 2 | 1 | N/A | N/A | 1 | 3 | 2 | 0 | N/A | N/A | 2 | 0 | N/A | N/A | 5 |
| | 2 | 3 | 1 | 1 | 1 | 2 | 2 | N/A | N/A | 2 | 5 | 1 | 0 | N/A | N/A | 1 | 1 | N/A | N/A | 9 |
| | 5 | 1 | 0 | 0 | 1 | 1 | 1 | N/A | N/A | 1 | 1 | 0 | 0 | N/A | N/A | 1 | 0 | N/A | N/A | 1 |
| D | 8 | 0 | 0 | 0 | 0 | 0 | 0 | N/A | N/A | 1 | 0 | 0 | 0 | N/A | N/A | 0 | 0 | N/A | N/A | 1 |
| E | 4 | 1 | 2 | 0 | 0 | 3 | 3 | N/A | N/A | 0 | 1 | 1 | 0 | N/A | N/A | 0 | 0 | N/A | N/A | 0 |
| Total | | 15 | 10 | 9 | 10 | 15 | 21 | | | 12 | 14 | 11 | 2 | | | 7 | 1 | | | 16 |

trees (S2 Fig). Genotype assignment into B66, B3 and B68 was possible with their REA pattern information.

A detailed temporal distribution of HAdV species and genotypes detection is described in Table 1. Initially, species B and C were predominant but since 2014 species B was no longer detected. Specifically, genotype B3 was detected from 2000 to 2014, genotype B66 from 2000 to 2005, and genotypes C1 and C2 were detected throughout the whole studied period. Genotype E4 was sporadically detected and D8 (respiratory associated) was only detected twice (in 2008 and then in 2018).

## HAdV genetic diversity

To assess the hexon genetic diversity, a total of 1485 HAdV sequences from GenBank were included as background (S1 Table). Maximum likelihood and Bayesian phylogenies (S3 and S4 Figs, respectively) obtained from 6 out of 9 datasets showed one main clade accounting for most of the sequences including those obtained in this study.

However, for three datasets (genotypes C2 and 5 and E4; Fig 2) secondary clusters were observed.

Within C2 (Fig 2I) two main clusters were observed: the major one (which includes the reference, and most of the sequences) and a secondary one, characterized by a codon duplication in position 150 of the hexon gene (E150dup) and an M305L substitution. Most of our sequences (22/27) were included in this secondary cluster. In addition, within this cluster, a supported Argentinean subcluster was observed (Fig 2I, AR Clade). This group was characterized by 3 non-synonymous substitutions: E to K as the previously described E150dup insertion, K171N and A225P and 3 synonymous substitutions (Fig 3).

For HAdV-C 5 (Fig 2II) the sequences were clustered in two groups of similar size. The grouping was consistent with a 7 amino acid signature IVEGQYG or MNDAGNV (corresponding to positions 144, 188 to 191, 212, and 287 of the hexon gene for AC_000008 reference sequence). One sequence of this study grouped with the HAdV-C 5 reference sequence (IVEGQYG signature) while the remaining 7 were included in the other cluster (MNDAGNV signature).

In the case of HAdV-E 4 (Fig 2III) the sequences formed two well-described subgenotype groups denominated 4p and 4a [30]. These groups were consistent with a 9 amino acid signature SDAGKPSTT for 4p and ANVDNTGNK for 4a (positions 134, 135, 147, 199, 214, 240,

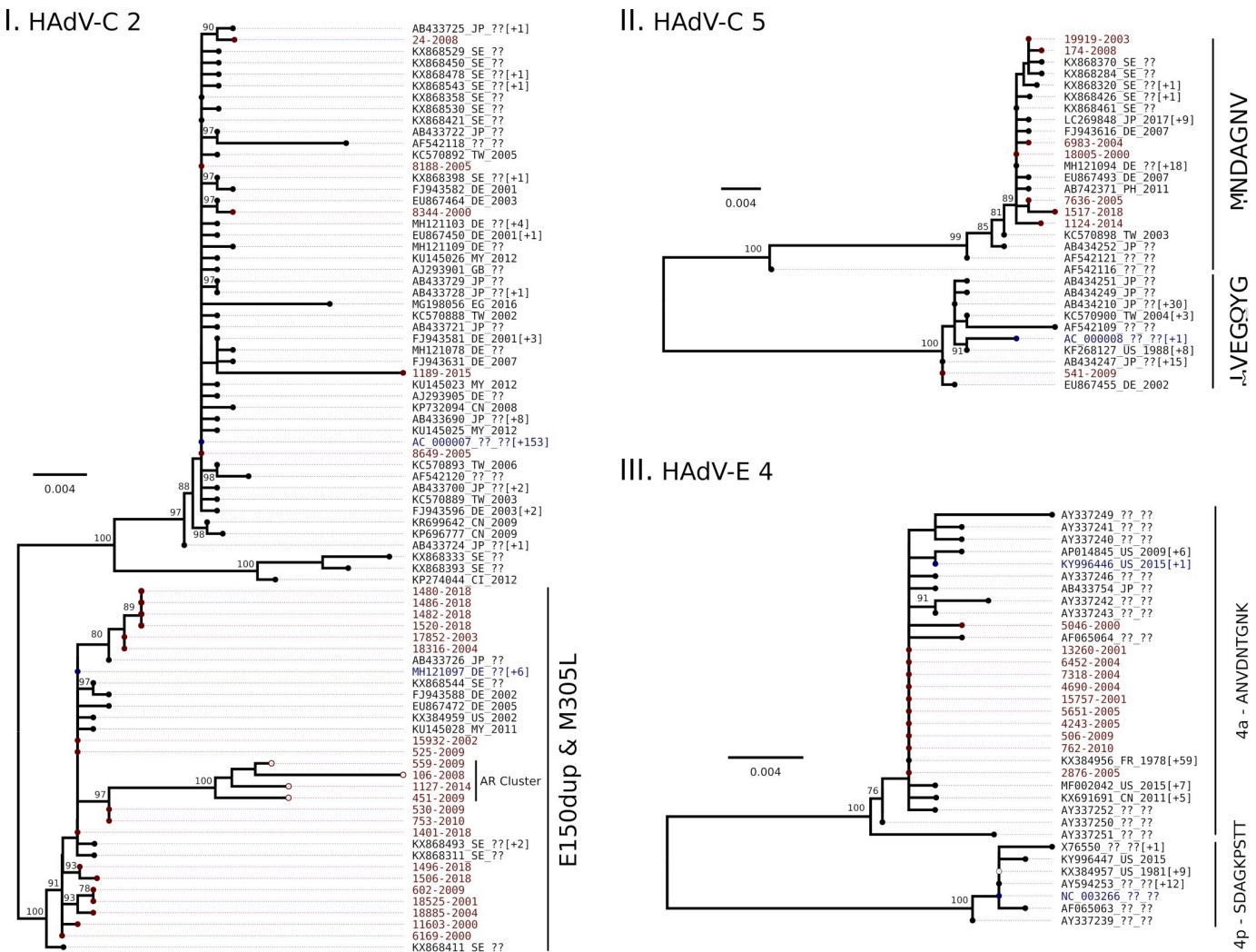

**Fig 2. Phylogenetic analyses of HAdV genotypes C2 (I), C5 (II) and E4 (III).** Maximum likelihood phylogenetic trees for sequences from Argentina (red), and from other countries (black). The number between square brackets indicates how many additional HAdV sequences had the same position in the complete tree. The reference sequence(s) for each genotype are in blue. The best fit model was used according to Bayesian Information Criterion (I: HKY+I, II and III: HKY+Γ) for each tree. The numbers near the node represent the branch support (bootstrap % over 1000 pseudoreplica). Amino acid signature or particular features are described at the right of each cluster. I. AR Clade: sequences with an amino acid substitution E to K in the duplicated codon. III. Hollow bullets: sequence with a 4 amino acid deletion at positions 192 to 195.

256, 258 and 289 of the hexon gene from the AY487947 HAdV-4 reference sequence). The clade for 4a was larger and included all the 11 samples from this study.

## Demographic and clinical features of HAdV infected children

Demographic, clinical characteristics and HAdV species distribution are detailed in Table 2. Briefly, the median age of the patients was 14.5 months (IQR: 7–23.5), 58.9% were male, and 12.1% were born preterm. Most of them (89.4%) were inpatients, 17 were admitted to the intensive care unit (15 required mechanical ventilation support). The most frequent clinical diagnosis (75.9%) was lower ARI. Most patients recovered (86.5%) from the respiratory disease, but 13 (9.2%) developed long-term lung sequelae and 6 (4.3%) died.

Patients infected with species E were older than those infected with other species (p = 0.013).

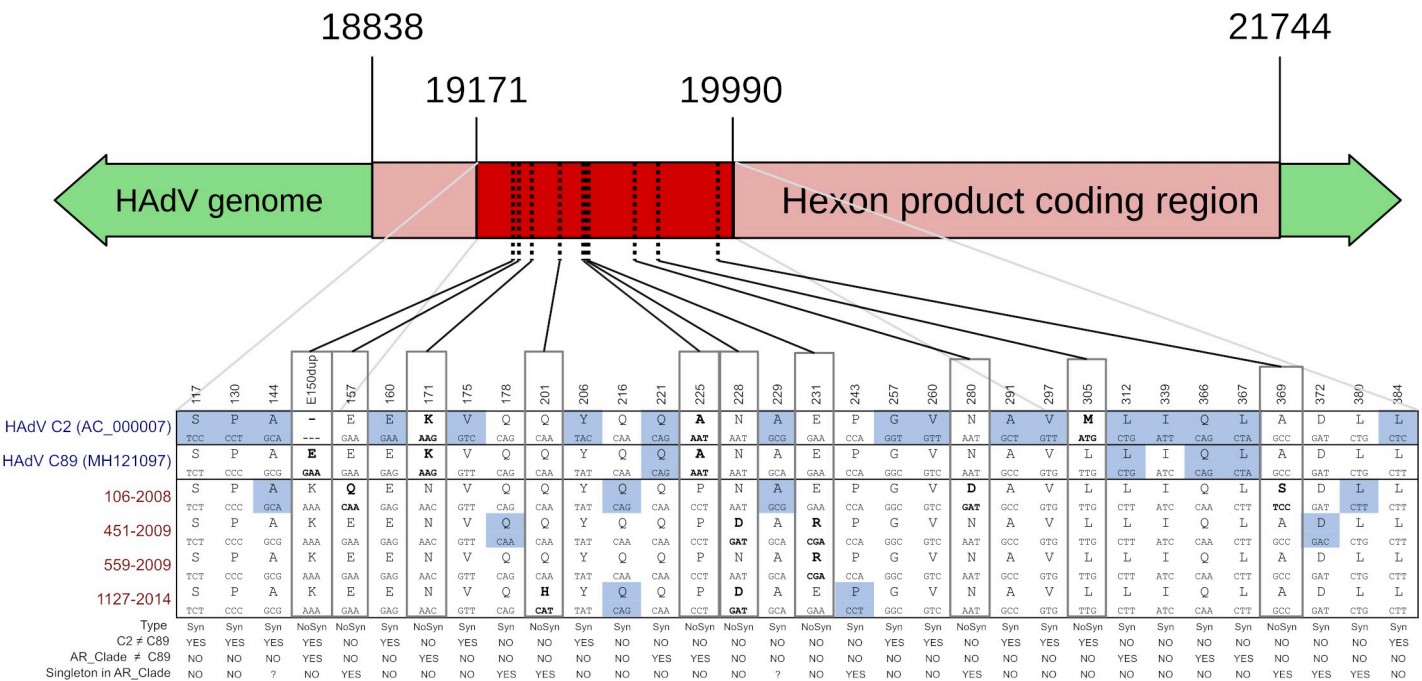

**Fig 3. Sequence differences between HAdV-C2 clusters.** Differences at the sequenced region for HAdV C2, C89 and AR Clade are shown. For each position, the upper letters represent one-letter amino acid codes, and below the nucleotide triplet that encodes it. Letters in bold representing non-synonymous substitutions, also shown as dotted lines in the genome. Letters with light blue background represent synonymous substitutions. Type: Syn for synonymous and NoSys for non-synonymous substitutions. Question marks indicate sites that are singletons in AR Clade but are shared with C2.

Being infected with species B had a higher risk for long-term sequelae or death compared to species C (OR: 4.17, CI95%: 1.02–17.07, p = 0.047), when adjusted by age, sex and prematurity.

Among the 13 children who developed long term lung sequelae, eleven (84.6%) were infected with species B (6 had B3, 4 had B66 and 1 had B55). Only two patients had species C (genotype C1).

All 6 fatal cases presented underlying clinical conditions: two had genetic diseases, one had hydrocephaly, one had encephalopathy, one had an underlying respiratory condition and one had a secondary immunodeficiency. Five out of 6 fatal cases were children less than 2 years old, two were premature and 4 were from low socioeconomic status. All of them had pneumonia, required mechanical ventilation support, and were hospitalized with a median length of stay of 30 days. Four patients were infected with species B (two B3 and two B66). The other two fatal cases had C1.

## Discussion

The molecular methods based on PCR and sequencing have replaced the classical serological typing methods in virology laboratories since classical assays require not only very well-trained and dedicated personnel but also the availability of a reference antisera panel for neutralization tests. As a quick and easy approach, the hexon gene sequencing has proved to be adequate for the initial typing of samples in a clinical setting because it highly correlates with the classical serological typing systems [14, 15]. However, phylogenetic analysis for genotyping requires a robust and comprehensive dataset. The construction of these datasets is complex and cumbersome due to the lack of standard practices for sequence submission to GenBank. Thus,

**Table 2. Demographic and clinical characteristics of patients with ARI and a positive HAdV diagnosis in Buenos Aires, Argentina.**

|  | Total | (141) | Species B | (71) | Species C | (57) | Species D | (2) | Species E | (11) |
|---|---|---|---|---|---|---|---|---|---|---|
|  | N | % | N | % | n | % | N | % | n | % |
| Age, months (IQR) | 14.5 | (7–23.5) | 12.5 | (7–28) | 14 | (7–21) | 10 | (1–19) | 45 | (20–65) |
| Sex male | 83 | 58.87 | 44 | 61.97 | 34 | 59.65 | 1 | 50.00 | 4 | 36.36 |
| Preterm born | 17 | 12.06 | 5 | 7.04 | 12 | 21.05 | 0 | 0.00 | 0 | 0.00 |
| Low socioeconomic status | 58 | 41.13 | 32 | 45.07 | 20 | 35.09 | 0 | 0.00 | 6 | 54.55 |
| Middle socioeconomic status | 83 | 58.87 | 39 | 54.93 | 37 | 64.91 | 2 | 100.0 | 5 | 45.45 |
| **Clinical manifestations** |  |  |  |  |  |  |  |  |  |  |
| Fever | 121 | 85.82 | 66 | 92.96 | 44 | 77.19 | 1 | 50.00 | 10 | 90.91 |
| Tachypnea | 82 | 58.16 | 52 | 73.24 | 27 | 47.37 | 0 | 0.00 | 3 | 27.27 |
| Cough | 104 | 73.76 | 58 | 81.69 | 39 | 68.42 | 0 | 0.00 | 7 | 63.64 |
| Wheezing | 21 | 14.89 | 6 | 8.45 | 15 | 26.32 | 0 | 0.00 | 0 | 0.00 |
| Vomiting | 31 | 21.99 | 16 | 22.54 | 13 | 22.81 | 0 | 0.00 | 2 | 18.18 |
| Diarrhea | 25 | 17.73 | 16 | 22.54 | 8 | 14.04 | 0 | 0.00 | 1 | 9.09 |
| Rhinitis | 91 | 64.54 | 47 | 66.20 | 34 | 59.65 | 0 | 0.00 | 10 | 90.91 |
| Conjunctivitis | 32 | 22.70 | 22 | 30.99 | 5 | 8.77 | 2 | 100.0 | 3 | 27.27 |
| **Clinical diagnosis** |  |  |  |  |  |  |  |  |  |  |
| Lower ARI | 107 | 75.89 | 60 | 84.51 | 40 | 70.18 | 1 | 50.00 | 6 | 54.55 |
| Pneumonia | 40 | 28.37 | 29 | 40.85 | 10 | 17.54 | 0 | 0.00 | 1 | 9.09 |
| Bronchiolitis | 41 | 29.08 | 20 | 28.17 | 19 | 33.33 | 0 | 0.00 | 2 | 18.18 |
| Bronchitis | 9 | 6.38 | 3 | 4.23 | 6 | 10.53 | 0 | 0.00 | 0 | 0.00 |
| Unspecified lower ARI | 17 | 12.06 | 8 | 11.27 | 5 | 8.77 | 1 | 50.00 | 3 | 27.27 |
| Upper ARI | 34 | 24.11 | 11 | 15.49 | 17 | 29.82 | 1 | 50.00 | 5 | 45.45 |
| Laryngitis | 2 | 1.42 | 1 | 1.41 | 0 | 0.00 | 0 | 0.00 | 1 | 9.09 |
| Common cold | 22 | 15.60 | 7 | 9.86 | 13 | 22.81 | 1 | 50.00 | 1 | 9.09 |
| Pharyngitis | 10 | 7.09 | 3 | 4.23 | 4 | 7.02 | 0 | 0.00 | 3 | 27.27 |
| Hospitalized | 126 | 89.36 | 66 | 92.96 | 48 | 84.21 | 2 | 100.00 | 10 | 90.91 |
| Long-term lung sequelae | 13 | 10.32 | 11 | 16.67 | 2 | 4.17 | 0 | 0.00 | 0 | 0.00 |
| Death | 6 | 4.76 | 4 | 6.06 | 2 | 4.17 | 0 | 0.00 | 0 | 0.00 |

sequences and metadata are sometimes incomplete, inaccurate, deficient and even with typing mistakes. In our study, background datasets were carefully constructed and reviewed several times to provide a useful context for the analysis.

In this study HAdV molecular typing and phylogenetic analysis were successfully performed in respiratory samples from 141 children with ARI sampled during a 14-year period in Buenos Aires, Argentina. HAdV was detected throughout the whole year and in all studied years showing no seasonal epidemic behavior as it was expected for this virus [3, 31, 32]. The circulation of 4 HAdV species (B, C, D and E) and 11 hexon genotypes was documented. As seen in studies carried out in other parts of the world, species B and C were the most frequently detected followed by species E and D [3, 4, 31–34].

Phylogenetic analyses showed that most of our HAdV strains were closely related to those described worldwide. However, for HAdV C2, C5 and E4 there was evidence of intra-genotype diversification supported by specific amino acid signature patterns or even by one single-codon duplication.

Most of the HAdV C2 sequences from Argentina presented a single codon insertion (E150dup) and a M305L substitution in the hexon gene suggesting that may belonged to the recently described genotype C89 [35]. Unfortunately, authors in that publication stated that C89 has a C2 hexon, so the best way to confirm this genotype assignment would be by

analyzing their unique penton gene sequence, which was not available for our analyzed samples. Additionally, four sequences within this cluster formed a separated and supported (100% bootstrapping) monophyletic group. At the protein level, the particular features shared by this secondary group were a single codon insertion (E150dup) and a M305L substitution in the hexon gene. These changes may suggest the circulation of a local lineage of the C89 genotype. Alternatively, these differences could account for a new genotype derived from C89 in the same way that C89 is derived from C2. Even the synonymous mutations may be related to shifts in codon usage as a particular adaptation to local hosts. Whole genome sequencing of these isolates may shed light on this issue.

Regarding HAdV C5, two clusters are described by phylogenetic analysis at the whole genome level [35], suggesting the existence of two variants. The small fragment of the hexon region sequenced in our study allowed to differentiated them into IVEGQYG or MNDAGNV signatures, being almost all samples from Argentina similar to the non-reference variant (MNDAGNV signature).

For HAdV E4, all samples in our study belonged to the 4a cluster, which is also the most commonly detected in USA, both in military and civilian populations [26]. Due to the relatively low number of cases studied it could be a sampling bias but since most sequences from the database also belong to this group it may also account for a greater geographical dispersion of 4a than 4p.

An association between HAdV species and disease severity has been previously described [16]. While species C is more frequently associated with mild respiratory infection [36], species B has been associated with higher severity [16]. In our study, children with ARI infected with species B showed a higher chance of worse outcome including long-term respiratory sequelae or even death compared to those infected with C. Genotypes B3 and B66 were detected in children with a worse outcome. However, half of these severe cases had B3 (6/13) and only a third had B66 (4/13), with a similar proportion rate observed throughout the study: 33/66 for B3 and 20/66 for B66. Due to our study design, the switch between B66 and B3 observed in 2005 could be attributed to changes in the source of the samples, but it may also suggest the existence of an intra-species genotype dynamic. Genotype B66 was initially described by REA in 1992 in Argentina as genome type 7h and was later classified as intermediate variant 7–3 [37]. After whole genome sequencing, this variant was finally proposed as genotype B66 (http://hadvwg.gmu.edu). This genotype B66 was previously associated with high mortality. Studies performed in Argentina from 1984 to 1992 described a fatal outcome from 28.6% to 34.5% in hospitalized children with HAdV positive respiratory infections [38–40]. Based on REA analysis, HAdV B66 (described in those works as genome type 7h) was later detected in Chile and Uruguay and even later in Japan and USA [8–10, 41]. In GenBank, it is possible to find sequences classified as 7h from other countries, although most of them are hexon-based only sequences which are identical between both genotypes (B7 and B66). Studies linking the pathogenicity and severity associated with HAdV B66 should carefully evaluate socioeconomic status and comorbidities in their inclusion criteria and the genotyping method, preferring whole genome analysis strategies.

Some of the limitations of this study include a relatively small number of patients and a halt in patients´ enrollment in some years. Also, we are aware that samples from two hospitals from Buenos Aires do not represent the actual diversity of the virus in the whole city nor even the country but they are enough to fulfill our aim for describing circulating HAdVs genotypes. The detection of 11 genotypes, including a possible new variant, in this probably under-represented sample accounts for an actual higher genetic diversity for the whole country. The number of cases studied also implies that the demographic information about the patients and possible associations between genotypes and illness outcomes were only provided as a

description of the studied population. Even though phylogenetic analyses were carried out with all accessible sequences, different search strategies produce different datasets. In addition, information from several countries is not available and genotype assignation is sometimes wrong or encoded in different ways by the authors. Even though our dataset was constructed as carefully as possible, the global genetic diversity of HAdV may still be underestimated.

## Conclusion

The phylogenetic analysis using partial hexon gene sequences was a useful strategy for typing circulating HAdV genotypes. Furthermore, it was useful for the identification of subgenotype variants for HAdV C5, E4 and C2. Most of the HAdV sequences obtained in this study were similar to those already described in other countries. However, the identification of an Argentinean cluster within C2, suggests the existence of a new variant and deserves further studies at a whole genome scale.

Finally, this study contributes to the knowledge of HAdV epidemiology and provides insights about HAdV species and genotypes circulation in hospitalized or outpatient children with ARI from 2000 to 2018 in Buenos Aires.

## Supporting information

**S1 Fig. Data acquisition and analysis flowchart.**
(PDF)

**S2 Fig. Phylogenetic tree (Bayesian inference) HAdV strains from children with acute respiratory infection in Buenos Aires, Argentina -2000 to 2018.**
(PDF)

**S3 Fig. Genotype specific maximum likelihood phylogenies of HAdV strains from this study.**
(DOCX)

**S4 Fig. Genotype specific Bayesian phylogenies of HAdV strains from this study.**
(DOCX)

**S1 Table. Additional HAdV sequences downloaded from GenBank for genotype diversity assessment.**
(XLSX)

## Acknowledgments

The authors want to thank Dr. Patricia Murtagh† and Dr. Santiago Vidaurreta for patient enrollment, Mergiory Labadie-Bracho for PCR assistance, Carmen Ricarte for technical support and Dr. Carolina Torres and Dr. Laura Mojsiejczuk for critical review of the manuscript.

## Author Contributions

**Conceptualization:** Débora N. Marcone, Andrés C. A. Culasso, Rodolfo H. Campos, Guadalupe Carballal, Marcela Echavarria.

**Data curation:** Débora N. Marcone, Andrés C. A. Culasso, Diana Viale.

**Formal analysis:** Débora N. Marcone, Andrés C. A. Culasso.

**Funding acquisition:** Marcela Echavarria.

**Investigation:** Débora N. Marcone, Andrés C. A. Culasso, Diana Viale, Guadalupe Carballal, Marcela Echavarria.

**Methodology:** Débora N. Marcone, Andrés C. A. Culasso, Noelia Reyes, Adriana Kajon.

**Project administration:** Marcela Echavarria.

**Resources:** Adriana Kajon, Marcela Echavarria.

**Supervision:** Rodolfo H. Campos, Marcela Echavarria.

**Writing – original draft:** Débora N. Marcone, Andrés C. A. Culasso, Rodolfo H. Campos, Guadalupe Carballal, Marcela Echavarria.

**Writing – review & editing:** Débora N. Marcone, Andrés C. A. Culasso, Noelia Reyes, Adriana Kajon, Diana Viale, Rodolfo H. Campos, Guadalupe Carballal, Marcela Echavarria.

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
