## [Decision Letter · Decision Letter 0]

22 Dec 2020

PONE-D-20-36189

Genotypes And Phylogenetic Analysis Of Adenovirus In Children With Respiratory Infection In Buenos Aires, Argentina (2000 - 2018)

PLOS ONE

Dear Dr. Echavarria,

Thank you for submitting your manuscript to PLOS ONE. After careful consideration, we feel that it has merit but does not fully meet PLOS ONE’s publication criteria as it currently stands. Therefore, we invite you to submit a revised version of the manuscript that addresses the points raised during the review process. Particularly, the questions and comments raised by Reviewer 2 should be addressed. 

We look forward to receiving your revised manuscript.

Kind regards,

Dong-Yan Jin

Academic Editor

PLOS ONE

Journal Requirements:

Reviewers' comments:

Reviewer's Responses to Questions

**Comments to the Author**

1. Is the manuscript technically sound, and do the data support the conclusions?

Reviewer #1: Yes

Reviewer #2: Partly

2. Has the statistical analysis been performed appropriately and rigorously? 

Reviewer #1: Yes

Reviewer #2: Yes

3. Have the authors made all data underlying the findings in their manuscript fully available?

Reviewer #1: No

Reviewer #2: Yes

4. Is the manuscript presented in an intelligible fashion and written in standard English?

Reviewer #1: Yes

Reviewer #2: Yes

5. Review Comments to the Author

Reviewer #1: The article entitled “Genotypes And Phylogenetic Analysis Of Adenovirus In Children With Respiratory Infection In Buenos Aires, Argentina (2000 - 2018)” by Débora N. Marcone et al. is a cross-sectional study using HAdV positive respiratory samples from children with ARI, in Buenos Aires. The topic of the article is interesting and medically important. The manuscript is well written. My comments to the authors are as follows:

Abstract: OK.

Introduction:

Line 44: Describe the common adenoviral and occasional adenoviral disease separately.

Methods:

It will be well understood for the reader if the whole method is shown with the help of a flowchart.

Line 111, 112: Explain the rationale of direct PCR-sequencing of L1 (HVR1-6) of the hexon.

Line: 135-37: REA enzyme profile can be added as a supplementary file if available.

Result: OK

Discussion and conclusion: OK

Reviewer #2: This manuscript reported genotypes and phylogenetic analysis of adenovirus in children with respiratory infection in Buenos Aires, Argentina (2000-2018). Authors performed phylogenetic analysis based on sequences from the hexon gene region obtained from 141 patients and confirmed the circulation of the four species (B, C, D and E) represented by 11 genotypes, with predominance in species B and C and shifts in their proportion in the studied period (2000 to 2018). The following are some comments and suggestions for the authors:

1. Did authors checked their sequences for possible recombination before phylogenetic analysis?

2.How long is the hexon gene fragment been used in the phylogenetic analysis? Please state this clearly.

3. It is advised to use second method (e.g. Bayesian inference) to construct another phylogenetic tree. Comparing the phylogenies constructed by different methods is essential to avoid systemic errors in evolutionary study.

4. The phylogenetic trees should be simplified. There are a lot of very similar sequences from the same region and/or years making the trees very large and very difficult to inspect. Too many very similar sequences did not benefit to the phylogenetic analysis but rather interfere the inspection of tree topology. I would suggest to omit some identical sequences if any.

5. Why did the authors not include 7b reference sequences since 7b is one of the major genotypes either?

6. How did the authors distinguish B7 and B66? As I see, they cannot be distinguished in the phylogenetic tree based on the hexon gene fragment.

6. PLOS authors have the option to publish the peer review history of their article (what does this mean?). If published, this will include your full peer review and any attached files.

Reviewer #1: No

Reviewer #2: No

---

## [Author Response · Author response to Decision Letter 0]

4 Feb 2021

Dear reviewers

In this document you will find the response to all your comments. To easy the reading, they were written in blue arranged in line with your questions (in black).

Additionally, we would like to reinforce that all data associated to this study are supplied as supporting material, published at Mendeley database and GenBank:

• Marcone, Debora; Culasso, Andres; Campos, Rodolfo; Echavarria, Marcela (2019), “ADENOVIRUS GENOTYPES TO BE USED IN PHYLOGENETIC ANALYSIS”, Mendeley Data, v1 http://dx.doi.org/10.17632/prf5vffgd2.1

• Culasso, Andrés; Marcone, Debora (2020), “Bibliographic Source for HAdV sequences”, Mendeley Data, V1, doi: 10.17632/jnh663g7w9.1 http://dx.doi.org/10.17632/jnh663g7w9.1

• Sequence data are available in GenBank under accessions numbers: MG000707 to MG000784 and MK913783 to MK913843.

Responses

• Reviewer #1: The article entitled “Genotypes And Phylogenetic Analysis Of Adenovirus In Children With Respiratory Infection In Buenos Aires, Argentina (2000 - 2018)” by Débora N. Marcone et al. is a cross-sectional study using HAdV positive respiratory samples from children with ARI, in Buenos Aires. The topic of the article is interesting and medically important. The manuscript is well written. My comments to the authors are as follows:

Abstract: OK.

Introduction:

Line 44: Describe the common adenoviral and occasional adenoviral disease separately.

The common and occasional adenoviral diseases were described separately

Methods:

It will be well understood for the reader if the whole method is shown with the help of a flowchart.

An additional supporting figure (Supporting Fig 1), containing a flowchart of data acquisition and analyses was added.

Line 111, 112: Explain the rationale of direct PCR-sequencing of L1 (HVR1-6) of the hexon.

Direct PCR-Sequencing of HVR 1-6 was used as a molecular alternative to neutralization assay as it was suggested by Lu and Erdman (2006). It represents a quick approach and requires few resources (laboratory and trained personnel). According to these authors, the analyzed region is thought to contain most of the antigenic determinants of the neutralization assays.

Line: 135-37: REA enzyme profile can be added as a supplementary file if available.

REA enzyme profiles as captured by photography are not available.

Result: OK

Discussion and conclusion: OK

• Reviewer #2: This manuscript reported genotypes and phylogenetic analysis of adenovirus in children with respiratory infection in Buenos Aires, Argentina (2000-2018). Authors performed phylogenetic analysis based on sequences from the hexon gene region obtained from 141 patients and confirmed the circulation of the four species (B, C, D and E) represented by 11 genotypes, with predominance in species B and C and shifts in their proportion in the studied period (2000 to 2018). The following are some comments and suggestions for the authors:

1. Did authors checked their sequences for possible recombination before phylogenetic analysis?

As usual before phylogenetic analyses, the presence of recombination was assessed. The use of traditional methods like RDP (recombination detect programs) is difficult due to the large number of possible parental genomes (+90) and the fact that some of the parentals (reference sequences) are recognized have a recombinant origin (for example HAdV B 16 and E 4). Additionally, the analyzed region is relatively small (~700 to ~800 depending on HAdV species and genotype) and it is not known to include recombination hotspot (that are located near the genomes ends). Our approach was to carefully examine the results of the FASTA36 local alignments made to classify the sequence to the species. In the case of a recombinant sequence, the program may inform secondary hits with different reference sequences. In our analyses, the only “multiple” hits results were in cases of well know shared hexon regions, like HAdV B 3 and 68, B 7 and 66, and C 2 and 89. Finally, in the phylogenetic analyses, no sequence was suspiciously divergent from the references, and genetic distances were too low to reliably test for recombination events.

2.How long is the hexon gene fragment been used in the phylogenetic analysis? Please state this clearly.

According to Lu and Erdman 2006, depending on the actual species and genotype the product of the nested PCR protocol should have between 688 and 821 bp. Information regarding the hexon gene fragment amplified and used in the phylogenetic analysis are now described in Methods (line 122). 

3. It is advised to use second method (e.g. Bayesian inference) to construct another phylogenetic tree. Comparing the phylogenies constructed by different methods is essential to avoid systemic errors in evolutionary study.

Supplementary phylogenetic trees using Bayesian inference were constructed, to be compared with those obtained using maximum likelihood, and presented as supporting fig 2 (genotype assignment) and supporting fig 4 (genetic diversity assessment). Although the topologies obtained with both methods are almost identical, differences in branch lengths were observed. Mainly, very short (but not insignificant branch lengths) were obtained in Bayesian methods while near-zero lengths were rounded up to zero in ML trees. Additionally, it should be note that due to the sequence similarity, the number of sequences and the star-like topology, the result of the Bayesian method should be taken with extreme care due to the Star Tree Paradox issue caused by the default priors on star-like topologies.

4. The phylogenetic trees should be simplified. There are a lot of very similar sequences from the same region and/or years making the trees very large and very difficult to inspect. Too many very similar sequences did not benefit to the phylogenetic analysis but rather interfere the inspection of tree topology. I would suggest to omit some identical sequences if any.

The fig 2 was now simplified. The full version of these trees is depicted in Supporting Figure 3 and 4, for ML and Bayesian method respectively. In the new version all the identical sequences from GenBank were represented by a single sequence and added the suffix “[+XXX]” to indicate the number of additional sequences that lied in the same position of the tree. However, all sequences from this study were included in this new version.

5. Why did the authors not include 7b reference sequences since 7b is one of the major genotypes either?

The reference sequence for 7b is now included in our reference dataset (http://dx.doi.org/10.17632/prf5vffgd2.1) and in the phylogenetic tree of figure 1. Originally, the reference sequence present in Harrach’s list was only those for 7a reference (a vaccine strain), we further included 7p (prototype: Gomen’s strain) and 7d2 (alternate vaccine strain). Since our REA results showed no 7b sequences we initially don’t include this reference sequence. At the analyzed region the 7b reference sequence is identical to 7a, 7d2 and 66.

6. How did the authors distinguish B7 and B66? As I see, they cannot be distinguished in the phylogenetic tree based on the hexon gene fragment.

Indeed, as stated in the previous response the sequences for B7 and B66 are identical at the hexon region. The fiver region of B66 is identical to B3 and thus, the antigenic / genetic differences are concentrated in the penton region. A similar situation is found between the hexon region of B3 and B68. Because of these, we now include a note in the line 169 of the revised manuscript explaining that the genotype assignment includes both, phylogenetic and REA results.

---

## [Decision Letter · Decision Letter 1]

22 Feb 2021

Genotypes And Phylogenetic Analysis Of Adenovirus In Children With Respiratory Infection In Buenos Aires, Argentina (2000 - 2018)

PONE-D-20-36189R1

Dear Dr. Echavarria,

We’re pleased to inform you that your manuscript has been judged scientifically suitable for publication and will be formally accepted for publication once it meets all outstanding technical requirements.

Kind regards,

Dong-Yan Jin

Academic Editor

PLOS ONE

Additional Editor Comments (optional):

Reviewers' comments:

Reviewer's Responses to Questions

**Comments to the Author**

1. If the authors have adequately addressed your comments raised in a previous round of review and you feel that this manuscript is now acceptable for publication, you may indicate that here to bypass the “Comments to the Author” section, enter your conflict of interest statement in the “Confidential to Editor” section, and submit your "Accept" recommendation.

Reviewer #1: All comments have been addressed

Reviewer #2: All comments have been addressed

2. Is the manuscript technically sound, and do the data support the conclusions?

Reviewer #1: Yes

Reviewer #2: Yes

3. Has the statistical analysis been performed appropriately and rigorously? 

Reviewer #1: Yes

Reviewer #2: (No Response)

4. Have the authors made all data underlying the findings in their manuscript fully available?

Reviewer #1: Yes

Reviewer #2: Yes

5. Is the manuscript presented in an intelligible fashion and written in standard English?

Reviewer #1: Yes

Reviewer #2: Yes

6. Review Comments to the Author

Reviewer #1: In their revised manuscript (PONE-D-20-36189_R1) entitled “Genotypes And Phylogenetic Analysis Of Adenovirus In Children With Respiratory Infection In Buenos Aires, Argentina (2000 - 2018)”, Débora N. Marcone et al., addressed the reviewers query meaningfully. The incorporation of reviewers' suggestions improved the manuscript a lot.

I have no further suggestion.

Reviewer #2: (No Response)

7. PLOS authors have the option to publish the peer review history of their article (what does this mean?). If published, this will include your full peer review and any attached files.

Reviewer #1: **Yes: **Arun Kumar Adhikary

Reviewer #2: No

---

## [Editor Report · Acceptance letter]

26 Feb 2021

PONE-D-20-36189R1 

Genotypes And Phylogenetic Analysis Of Adenovirus In Children With Respiratory Infection In Buenos Aires, Argentina (2000 - 2018) 

Dear Dr. Echavarria:

I'm pleased to inform you that your manuscript has been deemed suitable for publication in PLOS ONE. Congratulations! Your manuscript is now with our production department. 

Kind regards, 

on behalf of

Professor Dong-Yan Jin 

Academic Editor

PLOS ONE